# Sex Hormones in Lymphedema

**DOI:** 10.3390/cancers13030530

**Published:** 2021-01-30

**Authors:** Florent Morfoisse, Audrey Zamora, Emmanuelle Marchaud, Manon Nougue, Leila H. Diallo, Florian David, Emilie Roussel, Eric Lacazette, Anne-Catherine Prats, Florence Tatin, Barbara Garmy-Susini

**Affiliations:** UMR 1297-I2MC, Inserm, Université de Toulouse, UT3, 31432 Toulouse, France; florent.morfoisse@inserm.fr (F.M.); audrey.zamora@inserm.fr (A.Z.); emmanuelle.marchaud@inserm.fr (E.M.); manon.nougue@inserm.fr (M.N.); leila.diallo@inserm.fr (L.H.D.); florian.david@inserm.fr (F.D.); emilie.roussel@inserm.fr (E.R.); eric.lacazette@inserm.fr (E.L.); Anne-Catherine.Prats@inserm.fr (A.-C.P.); florence.tatin@inserm.fr (F.T.)

**Keywords:** lymphatic system, lymphedema, sex hormones, hormone therapy

## Abstract

**Simple Summary:**

Lymphedema is a life-long disease that affects a large number of patients treated for breast-, gynecological-, and urologic cancers in Western countries. Given that hormone levels are strongly modified in these conditions, and that patients widely undergo through hormone therapy, it is tempting to speculate that hormones might be key regulators in the maintenance of lymphedema. Despite an obvious prevalence for women, the role of sex hormones and gender has been poorly investigated in this pathology. This review aims to decipher how sex hormones interact with lymphatic vessels and whether hormone therapy could participate in lymphedema development.

**Abstract:**

Lymphedema is a disorder of the lymphatic vascular system characterized by impaired lymphatic return resulting in swelling of the extremities and accumulation of undrained interstitial fluid/lymph that results in fibrosis and adipose tissue deposition in the limb. Whereas it is clearly established that primary lymphedema is sex-linked with an average ratio of one male for three females, the role of female hormones, in particular estrogens, has been poorly explored. In addition, secondary lymphedema in Western countries affects mainly women who developed the pathology after breast cancer and undergo through hormone therapy up to five years after cancer surgery. Although lymphadenectomy is identified as a trigger factor, the effect of co-morbidities associated to lymphedema remains elusive, in particular, estrogen receptor antagonists or aromatase inhibitors. In addition, the role of sex hormones and gender has been poorly investigated in the etiology of the pathology. Therefore, this review aims to recapitulate the effect of sex hormones on the physiology of the lymphatic system and to investigate whetherhormone therapy could promote a lymphatic dysfunction leading to lymphedema.

## 1. Introduction

Sex hormones play a crucial role in physiology starting at the embryonic stages. Hormones exert a profound control over male and female biology and they influence the orientation of embryo development towards a male or female fetus. Then, they have a meaningful rebound in activity during puberty to promote gender-specific characteristics. Although sex hormones play a central role in reproduction, they also play a predominant role in many pathophysiological functions. In particular, for many years, the effect of sex hormones on cardiovascular function has been widely studied [1,2]. However, a large majority of preclinical studies have been performed on males due to the variations attributed to the sexual cycle in females, thus leading to a gap of comprehensive mechanisms related to gender.

The rise of cardiovascular diseases incidence and mortality with aging is known and established for decades, but it is intriguing to note that the sex is a major and universal risk factor for the occurrence of cardiovascular diseases since their incidence and mortality before the age of 50 are three times less in women than in men. Additionally, even if the difference related to gender decreases after menopause, women keep a lower incidence of coronary artery disease and myocardial infarction than men of comparable age. In that context, the vascular protective effect of estrogens is attributed to their receptor ERα expressed by the vascular endothelium [2]. Its ligand, 17β estradiol (E2), the most abundant estrogen, promotes endothelial cells proliferation, migration, survival, and vascular remodeling. Despite this important role in the physiology of the blood vascular endothelium, the role of E2 in the physiopathology of the lymphatic system has been poorly investigated. The lymphatic system is comprised of a network of blind-ended capillaries and collecting vessels, which drain interstitial fluids, fat, and immune cells through the lymph nodes to return them back to the blood circulation in the jugular area. It maintains fluid homeostasis, performs the immune surveillance and transports intestinal lipids [3]. Recently, study indicated that ERα controls the gene expression of lymphatic-related genes including VEGFR3, Lyve-1, and VEGFD [4]. Comparative studies of the lymphatic pumping pressure between males and females showed that the lymphatic pressure in females is lower than that of males, suggesting an influence of gender on the lymphatic pumping [5]. In the adulthood, lymphangiogenesis, the growth of new lymphatic vessels, occurs in pathological conditions including tumor metastasis, inflammatory, and cardiovascular diseases, whereas a dysfunction of the lymphatic system leads to lymphedema, an accumulation of fluids and adipose tissue in the limb [3]. Therefore, it is tempting to speculate that hormonal disorders involving E2 dysregulation may participate in lymphedema occurrence. Indeed, the deleterious effect of hormone therapy on the lymphatic endothelium [4] suggests that estrogen receptor antagonists can represent aggravating factors for lymphatic dysfunction such as lymphedema.

Lymphedema describes a progressive pathologic condition that arises as a consequence of impaired lymphatic drainage. This reduction of lymphatic functionality will result in interstitial accumulation of protein-rich fluid and subsequent inflammation, adipose tissue hypertrophy, and fibrosis in the affected area [6]. Lymphedema is a chronic, debilitating condition leading to complications including infection, functional disability, chronic cutaneous changes, and psychosocial morbidity [7]. Primary lymphedema is used to describe patients who have a congenital dysfunction in their lymphatic system caused by a genetic mutation in essential genes for lymphatic network formation (i.e., VEGFR3, FOXC2), whereas secondary lymphedema results from disruption or obstruction of a normal lymphatic system due to infection or caused by medical treatments (lymph nodes resection, cancer treatment). Secondary lymphedema appears months or even years after surgery, suggesting that it is not only a side effect of the surgery, but can be driven by cancer treatments. Radiation therapy causes damage to the lymphatic vessels [8]. Additionally, surgery, specifically the surgical removal of lymph nodes, is involved in the development of secondary lymphedema. In the past years, surgical procedures have been improved (i.e., sentinel lymph node dissection) and lower the risk of developing lymphedema. So far, most lymphedema studies have focused on lymphatic vessels neglecting their microenvironment, in particular, fibrosis that appears to be of paramount importance as it is closely associated with abnormal lymphatic vessel formation in lymphedema. Indeed, lymphatics depend on compliant tissue to remain functional but as lymphedema-associated fibrosis increases, lymphedematous tissue becomes denser, harder, and inflexible. This will snowball into greater obstruction of lymphatic circulation, which in turn worsens lymphedema. Therefore, the normalization of the lymphedematous fibrosis remains a crucial issue for lymphedema treatment. In that context, TGFβ plays a central role in tissue fibrosis and was described to inhibit lymphangiogenesis [9]. In the mouse tail lymphedema model, TGFβ increases lymphedema, suggesting a deleterious effect for lymphatic growth and repair [10]. However, in certain tissues and pathological conditions such as renal and peritoneal fibrosis, TGFβ signaling is coordinated with VEGF-C signaling to significantly upregulate lymphangiogenesis, leading to novel therapeutic strategies for lymphedema [11]. In addition, recent studies demonstrated that the formation of de novo lymphatic vessels considerably ameliorates the condition after adipose-derived stem cells transplant [12]. Ogino and colleagues demonstrated that formation of new lymphatic vessels via intussusceptive lymphangiogenesis, improved fibrosis, and dilatation capacity of the lymphatic vasculature that is necessary to restore the drainage in lymphedema. However, seventy to eighty percent of breast cancers are hormone-dependent and most women are treated during more than five years by hormone therapy following the surgical procedure [13]. This review aims at recapitulating the current knowledge on hormones and hormone therapy on lymphedema. To date, the lack of direct evidence of the role of hormone therapy in lymphedema is attributed to the absence of single-treatment, as combining hormone- and chemotherapy impairs the dissection of molecular mechanisms involved in the pathology. The major international guidelines recommend tamoxifen as adjuvant endocrine therapy for premenopausal women with hormone receptor-positive breast cancer, whereas aromatase inhibitor is predominantly given to post-menopausal women. Recent evidences suggest that tamoxifen modulates lymphangiogenic gene expression and participates to lymphatic dysfunction and leakage [4].

Lower extremity lymphedema also affects a large number of patients treated for gynecologic malignancies, with published rates as high as 70% in select populations [14]. Tamoxifen is the major drug for treating hormone-dependent breast cancer. However, ovarian cancers are known to possess receptors for estrogens and may thus also respond to tamoxifen. To our knowledge, there is no evidence to suggest hormone therapy would benefit patients with relapsed ovarian cancer and it is known to increase the risk of developing endometrial cancer and uterine cancer, in particular, in premenopausal women. Therefore, the incidence of lower limb lymphedema may not be associated with hormone treatment and might be restricted to surgery and radiotherapy in women.

In men, the development of secondary lymphedema has been observed in patients after prostate cancer treatments. It occurs after lymphadenectomy and radiotherapy to the pelvic lymph nodes, but the frequency remains low in patients with high-risk node-positive prostate cancer [15]. Despite a lack of evidence showing a direct link between male hormones and lymphatic function, we cannot exclude a role of hormones in the etiology of male lymphedema. This can be related to estrogens as their effects in physiology has been in an inaccurate way systematically attributed to women. However, men produce estrogens that can bind the estrogen receptors, which are wildly expressed in several tissues. In particular, both ER-α and ER-β are expressed in the normal prostate leading to the hypothesis that increased circulating estrogens might elevate prostate cancer risk. Additionally, the conversion of testosterone into 17β estradiol by the enzyme aromatase is the major source of estrogen in men, in particular in fat tissue [16]. Estrogens are clearly linked to the development and progression of prostate cancer and hormone therapy can provide benefit as second-line therapy. In addition, there is some evidence to suggest that antiestrogens can have therapeutic benefit as well, but this has not yet been explored clinically.

Therefore, it is urgent to further decipher the hormone-induced molecular mechanisms that control the lymphatic endothelium function in order to better understand the role of hormone vs. hormone therapy on the lymphatic dysfunction. This review aims to recapitulate the convergence of signs showing that sex hormones may represent a protective factor for the lymphatic system and that hormone therapy is a major risk factor for lymphedema that requires a specific follow up to significantly improve the quality of life of patients after cancer treatment.

## 2. Sex Hormones and Lymphatic Vasculature

### 2.1. Steroids Biosynthesis

Steroid biosynthesis starts with the conversion of cholesterol to pregnenolone [17]. Steroidogenic enzymes are responsible for the synthesis of various steroid hormones including glucocorticoids, mineralocorticoids, progestins, androgens, and estrogens (Figure 1). Whereas three endocrine organs, the adrenal gland, testis, and ovary are responsible for de novo steroid production, the biologically active steroids are also synthetized in numerous organs including brain, placenta, and adipose tissue. The lymph contributes to the systemic availability of hormones, but yet, only female hormones (i.e., progesterone and estrogen) have been identified as key players of the lymphatic physiology. Lymphatic endothelial cells (LEC) express both progesterone and estrogen receptors. Interestingly, progesterone receptor-positive endothelial cells were restricted to venules and lymphatics of the uterus and ovary, but absent from arterioles. Under physiological conditions, progesterone receptor promoter activity was not detected in the vascular beds of any other organs, revealing an exclusive organ-specificity for progesterone to the uterus and ovary [18]. Studies from Goddard and colleagues have shown that progesterone receptor expression in the endothelium is not constitutive. On average, at any given time, progesterone receptor positive endothelial cells represent more than 20 percent of uterine venous and lymphatic vessels. The expression in both vascular beds was significantly increased following hormone treatment [18]. However, there is very little data available about the role of progesterone on the lymphatic system in physiology and in pathology. To date, the majority of the literature has been focusing on the effect of estrogens and their receptor (ER), widely expressed in human tissues.

### 2.2. Source of Estrogen

Estrogen are steroids formed from androgen precursors, that are secreted mostly by the ovaries, placenta, and testes, and that stimulate the development of female secondary sex characteristics. Among the estrogens, E2 is the most potent estrogen and plays a role in almost all cells and tissues in the body [19]. Endocrine organs, in particular, adrenal gland and ovary produce estrogen. In the adrenal gland, the role of the lymphatic system is restricted to the transport of hormones. However, in ovaries, the lymphatic function is in part controlled by estrogens. Estrogens promote the opening of the initial lymphatics in the ovarian bursa to improve the fluid drainage that is necessary to maintain a suitable environment for ovulation [20].

On the other hand, in peripheral tissues, the origin of estrogen comes from the conversion of androstenedione produced by the adrenal gland by aromatase enzyme. Therefore, aromatase expression is highly regulated in a tissue–specific manner by alternative splicing and at the translational level via a specific 5′ mRNA untranslated region (5′UTR) [21]. For instance, the local source of estrogen is of particular importance in bone mineralization in male and female [22]. In skin, aromatase is mostly expressed by the stroma vascular fraction, mainly by fibroblasts located in dermis and in subcutaneous adipose tissue. Interestingly, aromatase activity varies between men and women, as a higher aromatase activity is detected in men sub-cutaneous adipose tissue [23]. However, estrogen stimulates adipose progenitor cell proliferation in a higher rate in women [24]. Current literature argues in favor of an anti-lipogenesis effect of estrogen associated to insulin sensitivity, glucose tolerance, and a reduction in white adipose tissue mass in both sexes [25]. However, response of pre-adipocytes to estrogen may differ between males and females, with a tendency of greater adipogenesis in females. This supports the idea that the adipose tissue microenvironment in females may be favorable to adipogenesis [26]. These studies highlight the potential role of “non-hormonal” tissues, mainly adipose tissue, as an important producer of estrogen which is of great importance in lymphedema as a strong accumulation of adipose tissue has been characterized in this pathology.

### 2.3. Estrogen Receptors

The biological effects of E2 are initiated after binding to the estrogen receptor α and β (ER-α and -β) [23], belonging to a nuclear receptor subfamily of ligand-inducible transcription factors [27,28]. Beside ERs, G protein-coupled estrogen receptor 1 (GPER), has emerged as a third ER. GPER activation has been reported to exert several beneficial effects in the cardiovascular system, including protection against atherosclerosis and hypertension in post-menopausal women [29]. Nevertheless, ERα, but not ERβ, mediates most of the vascular effects of estrogen, both on vascular and lymphatic vessels. The receptor alpha consists in a DNA binding domain (DBD), a ligand-binding domain (LBD), and two activation functions (AF1 and AF2) [30]. ER leads to enzymatic activities involved in chromatin remodeling, histone post-translational modifications, initiation or elongation of RNA transcription. Alternatively, ERs can also modulate gene expression without direct DNA binding, but through interaction with other transcription factors, such as AP1 or SP1, that mediate steroid signaling. Besides these nuclear actions, ERα can be localized at the plasma membrane, where it mediates rapid signaling effects that have been found in particular during endothelium healing [31]. In addition, the vascular effect of E2 has been in large part attributed to the production of endothelium-derived mediators, such as nitric oxide (NO) and anti-oxidative stress molecules including prostaglandin and cyclooxygenases derivatives [32,33,34,35]. In addition, E2 directly acts on the blood endothelium by modulating expression of vascular growth factors (VEGF-A, FGF2) and their receptors (VEGFR-2, FGFR-1) [36,37].

### 2.4. Protective Effect of Estrogen on Vascular and Lymphatic Vessels

A defect in estrogens has been associated with cardiovascular diseases as women develop two–three times less cardiovascular diseases than men before menopause [31]. Estrogen is recognized to have vasculoprotective and pro-angiogenic activities in several vascular beds including sub-cutaneous adipose tissue [38]. A protective role is reported on inflammation, vascular function, fibrosis, and oxidative stress in diseases related mice models such as atherogenic lesions and cardiac ischemia [39]. Deciphering the protective effect of estrogen in coronary heart diseases remains a major clinical issue as women have reduced risk compared to men before menopause [40]. The atheroprotective effect of estrogen has been studied in animal model as E2 prevents from atheroma in monkey [41] and in mice [42]. E2-vascular protective effect is in part due to its synergistic effect with endothelial growth factors VEGF-A [37] and FGF2 [36]. Estrogens improve endothelial repair after injury in large arteries such as carotid following balloon angioplasty and stents. Estradiol accelerates the reendothelialization by promoting endothelial proliferation and migration. This is mediated by its receptor ERα that drives arterial remodeling in response to ischemia and promotes an atheroprotective effect in mice models [43,44]. However, the effect of estrogens on large collecting lymphatic vessels remains unexplored.

Studies regarding the role of lymphatic vessels function in cardiovascular diseases are in their infancy. Consequently, the contribution of hormonal status in the onset of edema or lymphatic dysfunction is certainly underestimated. Yet, genomic effect of ERα regulates lymphangiogenic gene expression such as VEGFR-3, VEGF-D, and LYVE-1 [4]. Additionally, estrogen regulates hyaluronic acid production, a ligand of LYVE-1, through hyaluronan synthase transcription. Interestingly, Prox1, the master transcription factor of lymphatic endothelial cell identity, represses the transcriptional activity of the orphan estrogen-related receptor alpha (ERRα), therefore regulating a subset of genes implicated in metabolism and circadian rhythms in liver [45,46]. Prox1 is also known to regulate beta-oxidation during the specification of lymphatic lineage at embryonic stage [47]. Prox1 may play a pivotal role in the control of energy metabolism in several cellular subtypes. Although the orphan receptor ERRα does not bind the estrogen ligand, EERα and ERα regulate a fraction of common target genes including a subset of nuclear receptors [48]. Estrogens are also able to regulate key actors of lymphatic vasculature development such as adrenomedullin, its receptor RAMP3, and SVEP1 [49,50,51]. In addition, estrogen modulates the expression of SR-B1, a scavenger receptor involved in the transport of cholesterol from peripheral tissues to liver, in a tissue-specific manner [52]. Interestingly, lymphatic endothelium expresses HDL transporters such as SR-B1 and therefore participates in the reverse cholesterol transport [53]. Altogether, these studies suggest that estrogen may have a more broadly and unsuspected positive action on regulating lymphatic endothelium. Therefore, it would be of particular interest to further investigate their role in the context of lymphangiogenesis and lymphedema.

## 3. Sex Hormones and Lymphedema

Lymphedema is characterized by the abnormal swelling in a limb due to an alteration of lymphatic flow, accumulation of protein-rich fluid, and fibro-adipose tissue deposition. It can be an inherited condition (primary lymphedema) or occurs after cancer surgery and lymph node removal (secondary lymphedema). It is an unmet medical need affecting more than 250 million people worldwide and strongly impacting their quality of life [54].

### 3.1. Primary Lymphedema

Primary lymphedema represents only 10% of lymphedema cases worldwide and is mostly a pediatric pathology [55]. It was historically subdivided into two categories depending on the age of onset: congenital hereditary lymphedema (Milroy disease) occurs within the first two years of life while lymphedema praecox appears during puberty and represents the majority of primary lymphedema [55,56,57]. Recently, the St George’s classification algorithm has provided an accurate diagnosis for patients with lymphoedema based on age of onset, areas affected by swelling, and associated clinical features. Therefore, primary lymphedema can be divided in two subgroups including congenital onset (<1 year) and late onset (>1 year) [58]. Congenital onset is characterized by congenital unisegmental edema, lower limb lymphedema, and lower limb/genital edema. The late onset lymphedema is mainly associated with distichiasis syndrome (FOXC2 mutation). If not, it has to be classified into four-limbs or lower limbs only lymphedema [58]. Sexual differences are sharp in this pathology as primary lymphedema, and particularly late onset ones, develop preferentially in women with an average women to men ratio of 3:1 [59]. However, the role of hormones, in particular, estrogens, has been poorly investigated in primary lymphedema [55,59]. A large part of primary lymphedema is related to female hormone defect. In Turner syndrome, the abnormal ovarian development necessitates to provide estrogen supplementation to improve the patient condition [60,61]. However, the estrogen therapy has to be shortened to minimize the risk of endometrial cancer [62]. This resumes the eternal dilemma of the benefit versus risk on the use of hormones that are necessary to considerably improve the quality of life, but are also associated with the risk of developing hormone-dependent cancer and thus require to improve the patient follow-up. Another example is lymphedema-distichiasis which occurs at birth, however swelling often develops at puberty, suggesting a role of female hormones in the development of lymphedema [63].

Interestingly, a study focused on lymphedema inheritance reported that while women are more likely to develop lymphedema, the occurrence in men reflects a stronger genetic predisposition and a higher risk of hereditary transmission [64]. Given that primary lymphedema has been reported to be exacerbated at puberty and during menses and pregnancy, an implication of estrogens in lymphedema has been suspected. This is further supported considering that both E2 levels and lymphedema praecox occurrence rise between 11 and 14 years in women. E2 also downregulates the expression of Zona-ocluden 1 (ZO-1), a key component of LECs tight junction, which is thus expressed to a lesser extent in females [65]. Considering that the integrity of LEC junctions is critical to lymphatic drainage efficiency, E2-induced downregulation of ZO-1 may explain the increased susceptibility to lymphedema observed in women: during puberty, E2 level will rise and, through ZO-1 inhibition, increase the permeability of a lymphatic vasculature already hampered by a genetic mutation, thus triggering lymphedema development [66]. Yet, these data appear counter-intuitive since pro-lymphangiogenic roles of estrogens have also been identified. ERα activation following E2 stimulation is known to promote transcription of lymphangiogenic genes (VEGFR-3, VEGF-D, Lyve1) containing EREs [4]. Clinical studies also observed that women display higher endogenous serum levels of VEGF-C and -D than men [67]. Besides, E2 restricts water retention in tissues, suggesting a protective effect against lymphedema development [68]. One would thus expect that women with alteration in E2 metabolism may be more prone to lymphedema, but further investigations are needed to fully understand the mechanisms behind the increased susceptibility to primary lymphedema observed in women. Surprisingly, this sexual dichotomy is not observed for the congenital lymphedema that is developing before puberty [69]. Depending on the sex hormones contribution to their development, we can thus classify primary lymphedema in these two categories: congenital hormone-independent lymphedema and praecox hormone-dependent lymphedema.

### 3.2. Secondary Lymphedema

Secondary lymphedema develops as a consequence of disruption or obstruction of the lymphatic pathways and is usually divided into two categories: filariasis-induced lymphedema and iatrogenic ones.

#### 3.2.1. Filariasis Lymphedema

Filariasis is the major cause of lymphedema in the world and it is estimated that more than 200 million people are infected by *Wuchereria bancrofti*, *Brugia malayi*, and *Brugia timori,* the worms responsible for human filariasis [70]. Adult worms lodge in the lymphatic system, thus obstructing lymphatic vessels and disrupting lymphatic transport [71]. Like in primary lymphedema, there is a sexual dichotomy in lymphatic filariasis, but mostly in favor of females. Males of several mammary species have a higher prevalence of parasitic infections [72]. The influence of testosterone on this sex difference has been extensively studied. In their work, Nakanishi and colleagues reported that testosterone lowered the resistance to parasitic infection by inhibiting eosinophils activation (important effector cells for parasites) following parasites inoculation [73]. They were able to abolish the sex differences in infection susceptibility by treating female mice with testosterone prior to parasite exposition. Testosterone also displays immunosuppressive properties on T lymphocytes [74]. Interestingly, it has been suggested that gender dichotomy between males and females does not require continued presence of endogenous testosterone but rather depends on prepubertal testosterone pulses inducing physiological changes in males [75]. This is supported by the work of Ganley and Rajan who analyzed individual levels of testosterone in adult males and failed to find a correlation between serum levels of testosterone and susceptibility to infection [76]. Given that human epidemiological studies indicate a higher incidence of infection in men than in women associated with a higher worm burdens [77], deeper investigations are critical to understand the molecular mechanisms underlying this sexual dichotomy. This will allow a quicker identification of high-risk individuals among populations exposed to filariasis and a better adaptation of the therapeutic follow-up.

#### 3.2.2. Iatrogenic Lymphedema

In Western countries, lymphedema is a consequence of cancer treatments [78]. Nevertheless, genetic and environmental factors have also been proposed to participate in the outcome of the pathology. The most common example of iatrogenic secondary lymphedema is the upper limb lymphedema in women after lymph node dissection for breast cancer, and lower limb lymphedema following inguinal and pelvic lymph node dissection for pelvic neoplasms [79,80,81,82]. Currently, cancer-associated lymphedema prevalence is estimated to be the same in males and females, women representing the majority of patients being more likely due to the frequency of breast cancer by itself than to an increase susceptibility to secondary lymphedema. Nevertheless, our group demonstrated that estrogen effectively plays a role in lymphedema [4]. Using a murine model reproducing breast cancer-related lymphedema features, we observed a beneficial effect of estradiol on lymphatic function and drainage protecting the mice from edema formation. This effect is specifically mediated through ERα and is abolished when lymphedema is induced in Tie2-Cre; ERα^−/−^ mice in which the estrogen receptor is depleted in the endothelium. Interestingly, we observed that E2 stimulates both LEC migration and tubulogenesis, demonstrating for the first time a direct effect of estrogen on lymphatic endothelium (Figure 2). Considering that breast cancer-related lymphedema is the most frequent one and that hormone therapy is one of the main treatments used for this cancer, it is crucial to investigate if and how the different drugs used in this context, in particular, hormone therapies are implicated in lymphedema.

## 4. Hormone Therapy and Lymphedema

Hormone therapy is the gold standard therapy for estrogen receptor-positive breast cancer. It has been reported to have a beneficial effect on metabolic and vascular factors, influencing the incidence of coronary diseases in observational studies, but this aspect remains controversial among randomized clinical trials (Women Health Initiative, Heart and Estrogen/progestin Replacement Study, Estrogen Replacemend and Atherosclerosis trial) [40]. Thanks to the WHI study, we have observed an emergence of the scientific community interest in the study of the role of selective estrogen modulators (SERMs) in the cardiovascular system. However, SERMS exert an estrogen-agonist or -antagonist effect depending on the targeted tissue.

Approximately 70–80% of breast cancers are ERα-positive, and treated with endocrine therapies [1,83]. In this context, hormonal therapies for breast cancer are given after first confirming the expression of one of two hormonal receptors, ER and/or the progesterone receptor (PR). Depending on their hormonal status, women will be treated with either SERMs or aromatase inhibitors (AIs). SERMs are nonsteroidal compounds that function as ligands for ERs and act as agonists or antagonists in a target gene and in a tissue-specific fashion, whereas AIs block the conversion of androgens (testosterone, androstenedione) into estrogens (estrone and estradiol) [84,85,86].

Tamoxifen, the ER-partial agonist, is the most prescribed hormone therapy for premenopausal women with estrogen-dependent breast cancer since the seventies [87]. After menopause, women are given aromatase inhibitors that block the conversion of testosterone into estrogens to lower the level of circulating E2.

Tamoxifen has been used as a treatment for roughly four decades and has been approved as chemoprevention for over ten years [88]. Tamoxifen emerged as the first antiestrogenic agent that is clinically applicable to ER-positive patients with breast cancer [89]. Tamoxifen was initially developed to antagonize the tumor promoter effects of E2 by competing with it for binding to ER and was shown to effectively inhibit ER(+) breast cancer development in animal models and patients [89,90].

It was first designed to target mammary epithelial cells. However, its effect on the blood vasculature (i.e., coronary artery diseases) was rapidly investigated. Recently, our group has identified the expression of ERα in LECs [4]. We consequently postulated that women treated with SERMs and/or AIs could develop harmful side effects such as lymphedema. The correlation between hormone therapy and lymphedema is supported by the study of Das and colleagues performed on breast cancer survivors [91]. They found an increased risk of developing lymphedema in obese women treated with long-term tamoxifen and suggested the necessity of considering a weight reduction in these patients to improve the management of the pathology. Lower limb lymphedema was also found to be associated with tamoxifen treatment, which also contributes to the development of deep vein thrombosis [92]. Morfoisse and colleagues demonstrated the crucial role of female hormones, in particular, 17β-estradiol in maintaining the lymphatic function [4]. They identified the central role of ERα in the lymphatic endothelial function and highlighted for the first time in murine models a detrimental action of tamoxifen on lymphatic endothelial cells, leading to the development of lymphatic leakage and then lymphedema.

Tamoxifen treatment not only worsens lymphedema by inhibiting genomic action of ERα, but also blocks the membrane non-genomic activity of the receptor by inhibiting ERK phosphorylation, thus suppressing the protective effect of E2 when both treatments are administered together [4]. This study showed that women may develop more lymphedema after hormone therapy, and emphasized that the lymphatic vascular weakness leading to lymphedema is not only a side effect of surgery but also in part dependent on cancer treatment.

In addition, the aromatase inhibitors (e.g., anastrozole and exemestane) are used as a type of hormone therapy for postmenopausal women who have hormone-dependent breast cancer. To date, there is no rationale in the literature of a possible interference between aromatase inhibitors and lymphatic endothelium. In addition, in an experimental model of lymphedema, aromatase inhibitors had no effect on lymphatic leakage and leg swelling (unpublished data from Garmy-Susini’s lab [93]). Therefore, these data indicate that targeting ERα using tamoxifen seems to be more deleterious than blocking estrogen synthesis with aromatase inhibitors for lymphedema development (Figure 3).

## 5. Discussion and Future Perspectives

Although present at birth, vascular malformations more often develop during puberty. Unfortunately, these lesions may expand in adulthood and may lead to cardiovascular complications or decrease the quality of life. Nevertheless, a direct link between the level of estrogen in vascular and lymphatic malformations is lacking.

Most of the literature aiming to understand the molecular regulations of the LECs has been focused on the signaling pathways induced following the binding of lymphangiogenic vascular endothelial growth factors -C and -D (VEGF-C and VEGF-D) on their receptor VEGFR-3. VEGF-C promotes lymphatic endothelial repair in chronic diseases such as lymphedema, chronic inflammatory bowel diseases, and lung allograft [3].

In addition, VEGFC alone has appeared ineffective to improve lymphatic function in some mouse models of vascular injury, suggesting that it has to be combined to other molecules to fully restore the lymphatic function [94]. Sex hormones, and E2 in particular, might constitute potential partners for VEGF-C. In-depth studies of the mechanisms supporting sex differences in lymphedema susceptibility are still required before considering clinical applications. Interesting data have emerged from epidemiological studies showing that men are more affected by filariasis-related lymphedema, but the role of testosterone in this context remains elusive. It has been observed that testosterone inhibits eosinophil response to parasitic infection in mice models but this has not been confirmed in humans. Nevertheless, confirming in men the deleterious role of testosterone observed in mice and identifying how it disrupts the eosinophil response to parasite exposition may be a good strategy to design new treatments in order to suppress the exacerbated susceptibility in men and reduce the overall number of filariasis cases.

In Western countries, secondary lymphedema is most commonly due to cancer treatment. Fifteen to twenty percent of breast cancer patients suffer from lymphedema due to lymphadenectomy and/or radiation. Unilateral lymphedema occurs in up to 41% of patients after gynecologic cancer in which the hormone balance is strongly modified. Therefore, lymphedema has a high financial impact on national health systems. However, there is no cure and no medical treatment for cancer-related secondary lymphedema. The most common management of the pathology consists in a combination of manual compression, lymphatic massage, compression garments or bandaging that can only reduce transitorily the volume of the limb. Recently, surgical procedure has been improved to reduce lymphedema consisting in limiting lymph node excision to decrease the risk of developing the pathology [95]. However, despite the improvement of surgical technics, lymphedema remains a frequent concern of cancer patients.

This concern is further supported by the fact that lymphedema only develops in a subset of cancer patients who undergo lymph node dissection and do so in a delayed fashion, emerging often months and sometimes years after surgery. This suggest that additional factors play a role in lymphedema occurrence, but without prognostic markers currently identified, patients will live in fear of developing the condition for years. In breast cancer-related lymphedema, the hormone therapy emerges as a major risk factor. Tamoxifen is given during the 5 years following cancer surgery and has been associated to an increased risk of lymphedema development in clinical studies and suppressed lymphatic cell migration, thus worsening lymphedema in mice, while aromatase inhibitors do not hamper the lymphatic endothelium [96]. These data highlight the need to develop new SERMs without lymphatic adverse effects. They suggest to be more cautious with women who undergo tamoxifen treatment and to increase the medical follow-up. Considering the beneficial role of E2 on lymphatic vessels, an estrogen-based treatment might be worth considering at least for lymphedema following ER negative breast cancer. In this aspect, current studies on another estrogen-family member, the fetal estrogen estetrol (E4), will be interesting to assess the lymphatic properties of this promising molecule. E4 is described to have anti-tumor effects in patients with advanced ER^+^ breast cancer and could therefore represent a safe treatment for women who developed lymphedema after breast cancer. Future studies are required to achieve a more complete comprehension of how sex hormones affect lymphatic endothelial cell growth, function, and dysfunction in diseases. This will undoubtedly lead to exciting new data in the field, new treatment identification, and improved medical care for millions of lymphedema patients.

## 6. Conclusions

Sex hormones play a central role in the maintenance of the blood and lymphatic system. They consequently may have an important role in lymphatic disorders, in particular, in lymphedema. This review provides an overview of the clinical and molecular evidence showing that hormones must be taken in consideration for the future therapeutic orientations. It emphasized the necessity to be particularly vigilant with regard to the women who received long-term hormone therapy.

## Figures and Tables

**Figure 1 cancers-13-00530-f001:**
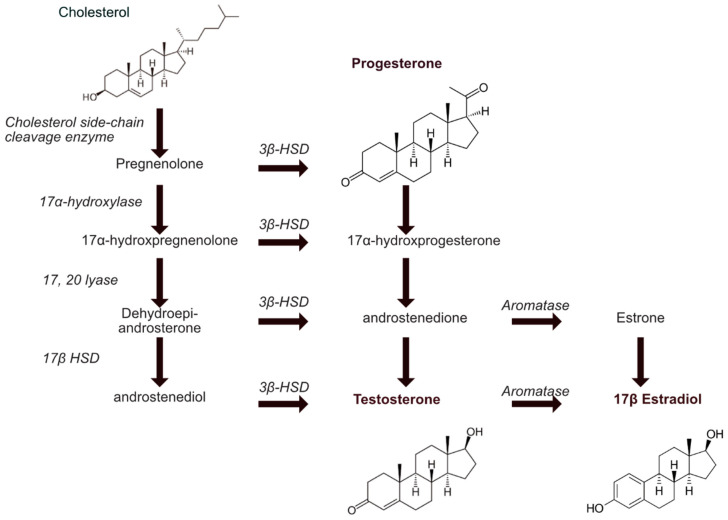
Diagram of sex hormones biosynthesis. Schematic representation of main sex hormones (in bold) biosynthesis starting from cholesterol. Enzymes responsible for each transformation step appear in italic. HSD: HydroxySteroid Dehydrogenase.

**Figure 2 cancers-13-00530-f002:**
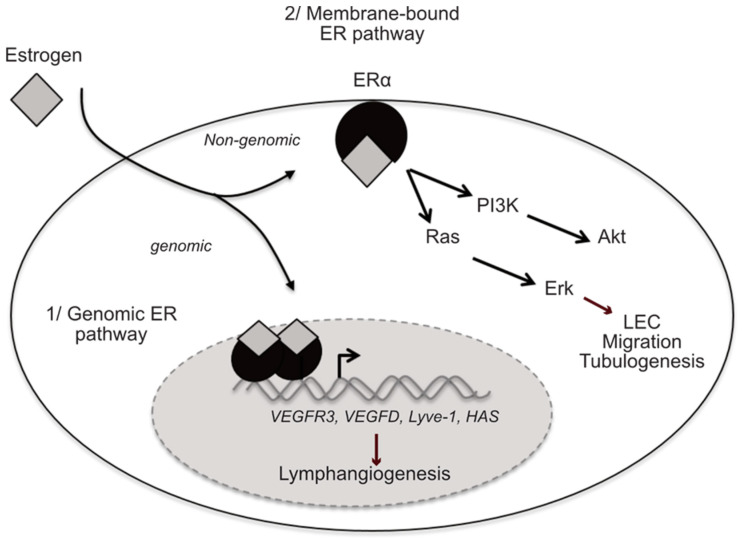
Estrogen-protective effect on lymphatic endothelium is mediated by the estrogen receptor (ER) α. Schematic representation of the molecular mechanisms implicated in the pro-lymphangiogenic effects of E2. The binding of E2 to ERα stimulates the transcription of lymphangiogenic genes through the genomic pathway. Additionally, E2 treatment activates Erk pathway and increases LEC migration and tubulogenesis.

**Figure 3 cancers-13-00530-f003:**
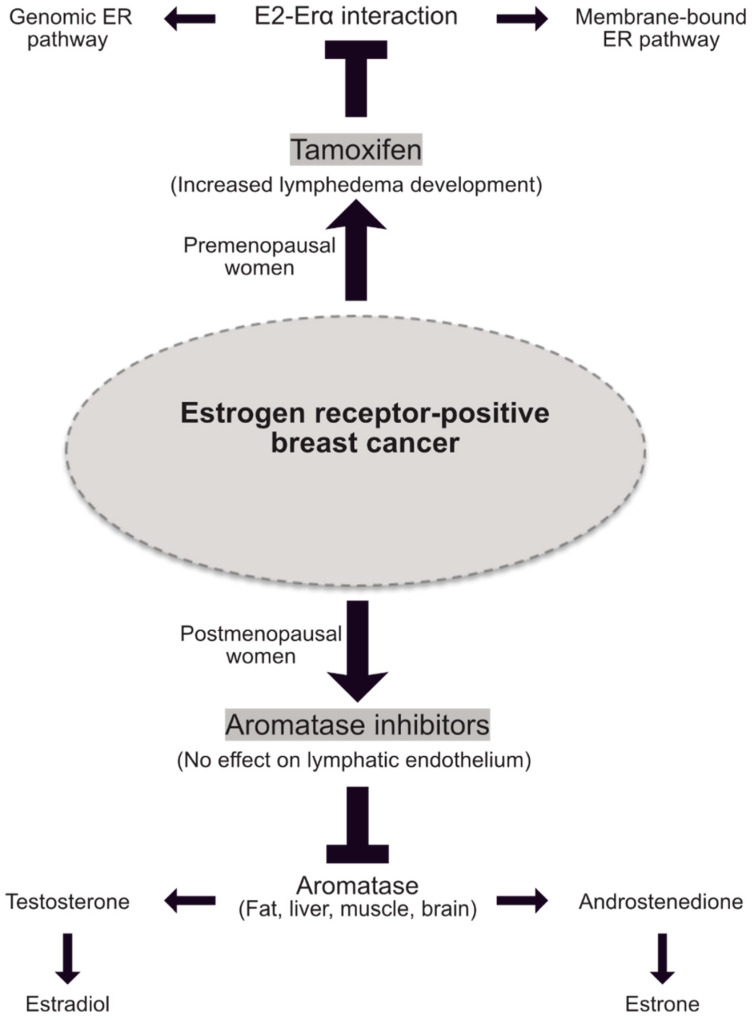
Hormone therapies in estrogen receptor-positive breast cancer. Schematic representation of the two main hormone therapy drugs used in breast cancer depending on the menopausal status of cancer patients. Tamoxifen abolishes E2-induced protective effect on lymphatic endothelium and thus participates to lymphedema development. At the contrary no effect on lymphatics has been reported for aromatase inhibitors.

## Data Availability

No new data were created or analyzed in this study. Data sharing is not applicable to this article.

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
