# Peer review of "Sex Hormones in Lymphedema"

_cancers, 2021, doi:10.3390/cancers13030530_

Round 1
Reviewer 1 Report
General view:
This review of sex hormones and lymphedema is much needed, and I am unaware of any other publications on this topic. However, the review is turgid with too much extraneous information making it difficult to read. It is too all encompassing and needs to focus down on the important areas where evidence exists. I suggest concentrating on lymphoedema secondary to cancer as this is where most evidence exists and will be of most interest to readers.
Other more specific comments:
- The classification of primary lymphoedema has changed. Because it persists into adulthood, most cases are adult.
- Turner’s syndrome needs discussion
- Filarial lymphoedema is not to my knowledge more common in females
Author Response
Reviewer 1
General view:
This review of sex hormones and lymphedema is much needed, and I am unaware of any other publications on this topic. However, the review is turgid with too much extraneous information making it difficult to read. It is too all encompassing and needs to focus down on the important areas where evidence exists. I suggest concentrating on lymphoedema secondary to cancer as this is where most evidence exists and will be of most interest to readers.
We agree with reviewer 1 as the role of sex hormones in lymphedema remains poorly explored. However, based on the fact that primary lymphedema develops preferentially in women with an average women-to-men ratio of 3:1, we estimated that it would deserve a paragraph to compile the scattered data of the literature.
Other more specific comments:
1. The classification of primary lymphoedema has changed. Because it persists into adulthood, most cases are adult.
We thank the reviewer 1 for pointing out this specificity of primary lymphedema. The main text has been completed based on st George’s classification algorithm of primary lymphatic anomalies (Kristiana Gordon 2020):
L254: “Recently, the St George’s classification algorithm has provided an accurate diagnosis for patients with lymphoedema based on age of onset, areas affected by swelling and associated clinical features. Therefore, primary lymphedema can be divided in two subgroups including congenital onset (<1year) and late onset (>1year). Congenital onset is characterized by congenital unisegmental edema, lower limb lymphedema, and lower limb/genital edema. The late onset lymphedema is mainly associated with distichiasis syndrome (FOXC2 mutation). If not, it has to be classified into four-limbs or lower limbs only lymphedema. (Kristiana Gordon 2020)”
2. Turner’s syndrome needs discussion
This part of the manuscript has been developed:
L266: “Thus, estrogen/progestin replacement therapy is required to achieve breast development and uterine maturation in Turner syndrome patients. However, the estrogen therapy has to be shortened to minimize the risk of endometrial cancer (Gravholt CH, 2016). This resumes the eternal dilemma of the benefit versus risk on the use of hormones that are necessary to considerably improve the quality of life, but are also associated with the risk of developing hormone-dependent cancer and thus require to improve the patient follow-up.”
3. Filarial lymphoedema is not to my knowledge more common in females
We agree with reviewer 1, the main text indeed shows that filarial lymphedema is more common in males:
L309 “Males of several mammary species have a higher prevalence of parasitic infections [60, 61]. The influence of testosterone on this sex difference has been extensively studied. In their work, Nakanishi and colleagues reported that testosterone lowered the resistance to parasitic infection by inhibiting eosinophils activation (important effector cells for parasites) following parasites inoculation [62].
Barbara Garmy-Susini

Reviewer 2 Report
The authors described "Sex hormones in lymphedema". There are few report regarding this topic. Actually, sex hormones may play an important role in lymphatic disorders. Women may develop more lymphedema after hormone therapy, and emphasized that the lymphatic vascular weakness leading to lymphedema is not only side effect of lymphatic dissection surgery but also in part dependent on cancer treatment including hormone therapy.
This article was well written and organized. However, I have some concerns and suggestion to improve the quality of this manuscript.
Major point
Lymphangiogenesis after surgery or radiotherapy is a key for the recovery of lymphedema. Recent articles regarding fibrosis related-lymphangiogensis should be added in Introduction (e.g. Kinashi H et al. IJMS 2018, Ogino R et al. IJMS 2020). Also, the main topic is intussucceptive lymphangiogenesis regarding the treatment of lymphedema. Please mention the treatment other therapies rather than hormone therapy in Introduction.
How about male after the surgery of prostatic cancer?
Minor point
- L155, "Protective effect of estrogen on vascular and lymphatic vessels" is a key point of this manuscript. Please more expand this section.
- L256, regarding cancer-associated lymphedema, How about lower limb lymphedema especially in gynecologic cancer patients? More references are needed.
- L367, references were lacking in mice studies.
Author Response
Reviewer 2
Comments and Suggestions for Authors
The authors described "Sex hormones in lymphedema". There are few report regarding this topic. Actually, sex hormones may play an important role in lymphatic disorders. Women may develop more lymphedema after hormone therapy, and emphasized that the lymphatic vascular weakness leading to lymphedema is not only side effect of lymphatic dissection surgery but also in part dependent on cancer treatment including hormone therapy.
This article was well written and organized. However, I have some concerns and suggestion to improve the quality of this manuscript.
Major point
Lymphangiogenesis after surgery or radiotherapy is a key for the recovery of lymphedema. Recent articles regarding fibrosis related-lymphangiogensis should be added in Introduction (e.g. Kinashi H et al. IJMS 2018, Ogino R et al. IJMS 2020). Also, the main topic is intussucceptive lymphangiogenesis regarding the treatment of lymphedema. Please mention the treatment other therapies rather than hormone therapy in Introduction.
We thank the reviewer 2 for his comments, the role of radiotherapy and fibrosis in lymphedema has been added to the manuscript as well as the selected references:
L76: Radiation therapy causes damage to the lymphatic vessels (Omar Allam, 2020). Additionally, surgery, specifically surgical removal of lymph nodes, is involved in the development of secondary lymphedema. In the past years, surgical procedures have been improved (i.e. sentinel lymph node dissection) and lower the risk of developing lymphedema.
L79: So far, most of lymphedema studies have focused on lymphatic vessels neglecting their microenvironment, in particular fibrosis that appears to be of paramount importance as it is closely associated with abnormal lymphatic vessel formation in lymphedema. Indeed, lymphatics depend on compliant tissue to remain functional but as lymphedema-associated fibrosis increases, lymphedematous tissue becomes denser, harder and inflexible. This will snowball into greater obstruction of lymphatic circulation, which in turn worsens lymphedema. Therefore, the normalization of the lymphedematous fibrosis remains a crucial issue for treatment of lymphedema. In that context, the TGFb plays a central role in tissue fibrosis and was described to inhibit lymphangiogenesis (Avraham, T.; 2010). In mouse tail lymphedema model, TGFb increases lymphedema suggesting a deleterious effect for lymphatic growth and repair (Clavin, N.W.; 2008). However, in certain tissues and pathological conditions such as renal and peritoneal fibrosis, TGFb signaling is coordinated with VEGF-C signaling to significantly upregulated lymphangiogenesis leading to novel therapeutic strategies for lymphedema (Kinashi H et al. IJMS 2018). Also, recent studies demonstrated that the formation of de novo lymphatic vessels considerably ameliorates the condition after adipose-derived stem cells transplant (Ogino R et al. IJMS 2020). The authors demonstrated that formation of new lymphatic vessels via intussusceptive lymphangiogenesis improved fibrosis and dilatation capacity of the lymphatic vasculature that is necessary to restore the drainage in lymphedema.
How about male after the surgery of prostatic cancer?
We thank the reviewer 2 for pointing out prostatic cancer-related lymphedema. As requested, we add the following paragraph in our manuscript:
L115: “The development of secondary lymphedema has been also observed in patients after prostate cancer treatment. It occurs after lymphadenectomy and radiotherapy to the pelvic lymph nodes, but the frequency remains low in patients with high-risk node-positive prostate cancer (Elisabeth Rasmusson 2013). Despite a lack of evidence showing a direct link between male hormones and lymphatic function, we cannot exclude a role of hormones in the etiology of male lymphedema. This can be related to estrogens as their effects in physiology has been in an inaccurate way systematically attributed to women. However, men produce estrogens that can bind the estrogen receptors, which are wildly expressed in several tissues. In particular both ER-α and ER-β are expressed in the normal prostate leading to the hypothesis that increased circulating estrogens might elevate prostate cancer risk. Also, the conversion of testosterone into 17β estradiol by the enzyme aromatase is the major source of estrogen in men, in particular in fat tissue (Maarten C Bosland,2005). Estrogens are clearly linked to the development and progression of prostate cancer and hormone therapy can provide benefit as second-line therapy. In addition, there is some evidence to suggest that antiestrogens can have therapeutic benefit as well, but this has not yet been explored clinically.”
Minor point
1. L155, "Protective effect of estrogen on vascular and lymphatic vessels" is a key point of this manuscript. Please more expand this section.
We expanded this section in the manuscript and added references about the protective effect of estrogen on vascular vessels.
L215: Estrogens improve endothelial repair after injury in large arteries such as carotid or carotid arteries following balloon angioplasty and stents. Estradiol accelerates the reendothelialization by promoting endothelial proliferation and migration. This is mediated by its receptor ERα that drives arterial remodeling in response to ischemia and promotes an atheroprotective effect in mice models (Billon-Gales, 2009; Gourdy, P.2007). However, the effect of estrogens on large collecting lymphatic vessels remains unexplored.
2. L256, regarding cancer-associated lymphedema, How about lower limb lymphedema especially in gynecologic cancer patients? More references are needed.
We expanded this section in the manuscript and added references about lower limb lymphedema in gynecologic cancer patients.
L107: Lower extremity lymphedema also affects a large number of patients treated for gynecologic malignancies, with published rates as high as 70% in select populations (Dessources K. 2020).
Tamoxifen is the major drug for treating hormone-dependent breast cancer. However, ovarian cancers are known to possess receptors for estrogens and may thus also respond to tamoxifen. To our knowledge, there is no evidence to suggest hormone therapy would benefit patients with relapsed ovarian cancer and it is known to increase the risk of developing endometrial cancer and uterine cancer, in particular in premenopausal women. Therefore, the incidence of lower limb lymphedema may not be associated with hormone treatment and might be restricted to surgery and radiotherapy.
3. L367, references were lacking in mice studies.
References have been added in the manuscript according to the reviewer request.
“Tamoxifen is given during the 5 years following cancer surgery and has been associated to an increased risk of lymphedema development in clinical studies and suppressed lymphatic cell migration thus worsening lymphedema in mice, while aromatase inhibitors do not hamper the lymphatic endothelium.
Barbara Garmy-Susini

Round 2
Reviewer 1 Report
No comments to authors
Reviewer 2 Report
The authors revised the manuscript precisely. I recommend Accept.